# The Dual Role of Myeloperoxidase in Immune Response

**DOI:** 10.3390/ijms21218057

**Published:** 2020-10-29

**Authors:** Jürgen Arnhold

**Affiliations:** Institute of Medical Physics and Biophysics, Medical Faculty, Leipzig University, 04 107 Leipzig, Germany; juergen.arnhold@medizin.uni-leipzig.de

**Keywords:** myeloperoxidase, neutrophils, immune response, phagosomes, cardiovascular diseases, chronic inflammation

## Abstract

The heme protein myeloperoxidase (MPO) is a major constituent of neutrophils. As a key mediator of the innate immune system, neutrophils are rapidly recruited to inflammatory sites, where they recognize, phagocytose, and inactivate foreign microorganisms. In the newly formed phagosomes, MPO is involved in the creation and maintenance of an alkaline milieu, which is optimal in combatting microbes. Myeloperoxidase is also a key component in neutrophil extracellular traps. These helpful properties are contrasted by the release of MPO and other neutrophil constituents from necrotic cells or as a result of frustrated phagocytosis. Although MPO is inactivated by the plasma protein ceruloplasmin, it can interact with negatively charged components of serum and the extracellular matrix. In cardiovascular diseases and many other disease scenarios, active MPO and MPO-modified targets are present in atherosclerotic lesions and other disease-specific locations. This implies an involvement of neutrophils, MPO, and other neutrophil products in pathogenesis mechanisms. This review critically reflects on the beneficial and harmful functions of MPO against the background of immune response.

## 1. Immune Response and Tissue Destruction

In humans and higher animals, protection against different threats that affect the homeostasis of host’s tissues is ensured by a coordinated action of the immune system in close association with activation of components of the acute phase, complement, coagulation, and contact systems [1,2]. An immune response starts usually with the recruitment and activation of innate immune cells such as neutrophils, monocytes and others, and can be followed by the additional activation of dendritic cells and lymphocytes. Importantly, the immune system is involved to resist the invasion of viruses, bacteria, fungi and other pathogens, as well as to remove and replace damaged material.

With these activities, immune response is primarily directed to diminish the degree of tissue destruction and to restore homeostatic conditions. Otherwise, immune cells themselves release numerous hazardous substances during their activation. These agents are predominantly used to inactivate, kill, and destroy pathogenic microorganisms. However, they can also damage cell material, cause necrotic cell death, and induce the release of antigens in the host. Altogether, the immune system plays a dual role with respect to cell and tissue damage [1]. On the one hand, it protects cells and tissues from many destroying threats. On the other hand, it can provoke significant harm to host tissues.

This general dual activity of the immune system is also reflected in the functional responses of many cells activated after their recruitment to inflammatory loci. Polymorphonuclear leukocytes (PMNs), also known as neutrophils, are key mediator cells of innate immunity. These cells are rapidly recruited to inflammatory sites. The infiltration of PMNs into infected and injured tissue is mediated by adhesion molecules, cytokines, chemotactic agents, and components of the extracellular matrix [3,4]. In this way, through endothelium and adjacent tissue, neutrophils are stepwise activated and can phagocytose foreign microorganisms at the destination site [5]. With the release of proteases, hydrolases, bactericidal proteins and others, and the generation of reactive species, PMNs contribute to the inactivation and killing of bacteria and fungi. Otherwise, these neutrophil-derived agents are also known to be able to damage intact cells and tissues.

The heme protein myeloperoxidase (MPO) is a major active component in activated human neutrophils [6,7]. It is also present in monocytes to a lesser extent, but usually lost during the maturation of these cells to macrophages [8,9]. Some reports describe MPO-rich macrophage subpopulations in atherosclerotic plaques, circulating blood, and multiple sclerotic lesions [10,11,12,13]. In macrophages, MPO can also result from phagocytosed apoptotic and necrotic PMNs or internalization of MPO-containing extracellular traps [14,15].

In this review, the major focus is on the characteristics of neutrophil-derived MPO. The main properties of MPO and physiological consequences of MPO activation are summarized during immune response. This concerns the role of MPO in neutrophil functions during inactivation and killing of ingested microbes and fungi as well as the potential contribution of released MPO in the modification of proteins in disease progression. In sum, this heme protein plays also a dual role in immune response with both helpful and devastating properties. A hypothesis about the switch from the protecting to harmful functions of MPO, neutrophils, and general immune functions is presented too.

## 2. Short Overview about Myeloperoxidase Properties

### 2.1. Selected Structural Properties

To discuss the role of MPO in immune response, it is first necessary to consider major characteristics of this enzyme. As there are numerous comprehensive reviews about structural, biochemical, and redox properties of MPO [6,7,16,17], I will here mention only the main issues.

Human myeloperoxidase is a cationic, dimeric protein with a ferric protoporphyrin IX (heme) in each subunit. In MPO, both heme groups are structurally and functionally identical [18]. Each heme is associated with the surrounding apoprotein by three covalent bonds, hence the heme is slightly bow-shaped along the axis from pyrrole ring A to ring C, with a shift in the central iron ion by 0.2 Å to the proximal side [19]. The three apoprotein-heme linkages determine the extraordinary biochemical and redox properties of this heme protein.

In each subunit of MPO, the heme is deeply buried into the protein bulk and connected with the surface by a narrow substrate channel. At the entrance to the distal cavity, there is a conserved hydrophobic region among mammalian heme peroxidases. Aromatic molecules and some other substrates can get close enough to the heme to allow an electron to transfer between the heme and substrates [16].

### 2.2. Heme States and Redox Properties of Myeloperoxidase

In resting MPO, the heme iron is in the ferric state (Fe^3+^). During activation, other heme states can result known as Compound I, Compound II, and Compound III (Table 1). Compounds I and II contain ferryl iron with an attached oxygen [16]. In Compound I of MPO, the porphyrin ring is additionally modified to a π-cation radical (^•+^Por) [20]. In Compound III, a superoxide anion radical (O_2_^•−^) is attached to ferric heme iron. This compound can also be presented as a resonance structure where dioxygen is bound to ferrous heme iron [21].

The oxidation states of Compounds I and II are higher by two or one units, respectively, in comparison to resting MPO (Table 1). Standard reduction potentials (*E*’°) for the interconversion of MPO states (Table 2) have been determined [22,23]. In biological systems, the conversion of resting MPO into Compound I, which plays a central role in reactions of the halogenation and peroxidase cycles (see below), is possible by the concurrent reduction of hydrogen peroxide to water.

### 2.3. Reaction Cycles of Myeloperoxidase

*Halogenation cycle:* An overview of the major reaction cycles of MPO is given in Figure 1. Compound I of MPO, which is formed as a result of a reaction between resting MPO and H_2_O_2_, is a short-lived, very reactive state [16]. It interacts rapidly by abstracting two electrons with (pseudo)halides and oxidizes them to hypohalous acids or hypothiocyanite. In this reaction, resting MPO is formed again. The reaction sequence resting MPO→Compound I→resting MPO comprises the halogenation cycle (see Figure 1). The standard reduction potential of the couple Compound I/resting MPO is known to be 1.16 V at pH 7 [24]. The reactivity of Compound I with (pseudo)halides decreases in the order SCN^−^ > I^−^ > Br^−^ > Cl^−^. This reactivity is also higher in slightly acidic media in contrast to neutral pH values [26].

On the basis of second-order rate constants and (pseudo)halide ion concentrations, it has been calculated that Cl^−^ and SCN^−^ are equally oxidized by activated MPO in a milieu that corresponds to pH and ion composition with 0.1 M Cl^−^ and 100 µM SCN^−^ of blood [27]. In secretions characterized by high-micromolar to low-millimolar SCN^−^ concentrations and the presence of the heme protein lactoperoxidase [28,29,30], MPO from recruited neutrophils and lactoperoxidase can together contribute to the prevailing formation of hypothiocyanite [31,32].

*Hypochlorous acid as a myeloperoxidase product:* In numerous reports, the formation of HOCl and chlorinated products by MPO was measured in the presence of taurine at pH 7.4 [33,34,35,36,37]. Other investigations did not detect chlorinated products in the absence of taurine at this pH [38,39,40]. Moreover, investigation of pH dependencies of the reduction potentials of involved redox reactions revealed that the probability of HOCl formation is very low at neutral pH values [40]. Interaction between Compound I and chloride yields a reversible high-spin complex [41]. Taurine is known to react with this complex to yield taurine chloramine and resting MPO without the formation of free HOCl [42,43].

In the cytoplasm of PMNs, the taurine concentration is 22–26 mM [44]. In plasma, taurine concentration is much lower, with about 38–40 µM [45]. In resting PMNs, both MPO and taurine are well separated. The formation of taurine chloramine inside neutrophils should be possible after permeabilization of azurophilic granules [46,47] or after leakage of phagosomes. It remains unknown whether MPO and taurine can interact together already immediately after the initiation of phagosome formation.

Numerous substrates are targets for free HOCl [48,49], and similarly for free HOBr [50]. The reactivity of OSCN^−^/HOSCN is more restricted and mainly concerns the oxidation of sulfhydryls [51]. As a result, HOSCN is much better able to penetrate into intact cells than HOCl and HOBr, and to thus exhibit a cytotoxic or bactericidal activity by affecting intracellular glutathione and other critical sulfhydryls [52,53]. Moreover, tyrosine residues of albumin are brominated by the MPO-H_2_O_2_-halide system in the presence of physiological concentrations of chloride and bromide at pH values higher than 7 [54,55].

*Peroxidase cycle:* Compound I is also able to oxidize numerous small substrates in a one-electron reaction under the formation of a substrate radical and reduction to Compound II. Important substrates are urate, tyrosine, tryptophan, sulfhydryls, nitric oxide, polyphenols, nitrite, xenobiotics, hydrogen peroxide, xenobiotics, and others [56,57,58,59,60].

The reduction of Compound II to resting MPO is more restricted due to a lower standard reduction potential for the couple Compound II/resting MPO (0.97 V at pH 7) in contrast to the couple Compound I/Compound II (1.35 V at pH 7) [25]). In tissues, only few substrates are known able to reduce Compound II at a sufficient rate. Among these substrates are polyphenols (such as (−)-epicatechin, luteolin, quercetin, (±)-eriodyctiol, and (+)-taxifolin [61,62,63], urate [60], ascorbic acid [64], tyrosine [56], serotonin [65], and superoxide anion radicals [66]. The highest rate for Compound II reduction by far was determined for quercetin, followed by (−)-epicatechin [61,63]. Compound II is unable to oxidize halides [26].

The reaction sequence resting MPO→Compound I→Compound II→resting MPO comprises the peroxidase cycle (see Figure 1). In the absence of substrates able to reduce Compound II, the latter compound can accumulate [7,67,68]. In summary, the reactions of the halogenating and peroxidase cycles of MPO are driven by substrate availability, reaction rates, and redox properties.

*Other reaction cycles:* Compound III is formed in a reaction of ferric MPO with a superoxide anion radical with a rate constant of 2 × 10^6^ M^−1^s^−1^ at pH 7.4 [69]. This rate constant is one order of magnitude higher than the rate constant for spontaneous dismutation of O_2_^•−^ at this pH [70]. In a reaction of Compound III with a further O_2_^•−^, which precedes with a rate of 1.3 × 10^5^ M^−1^s^−1^ at pH 7.4, resting MPO is recovered under the formation of H_2_O_2_ and O_2_ [66]. Compound III of MPO is not involved in the oxidation of halides and small substrates.

With further increasing pH values, all O_2_^•−^-dependent reactions of MPO become more likely, as the rate constant of spontaneous dismutation of O_2_^•−^ decreases steadily with increasing pH. These values are 6 × 10^4^ M^−1^s^−1^ at pH 8, 6 × 10^3^ M^−1^s^−1^ at pH 9, and 6 × 10^2^ M^−1^s^−1^ at pH 10 [70]. Thus, MPO may exhibit a weak superoxide dismutase activity at alkaline pH values. In this O_2_^•−^-driven reaction sequence, only resting MPO and Compound III are involved (see Figure 1).

At inflammatory sites, a very rapid reaction between NO and O_2_^•−^ may occur under the formation of peroxynitrite [71,72]. This reactive species is known to convert resting MPO directly to Compound II, whereby MPO may act as a sink for peroxynitrite [73]. The additional presence of substrates like O_2_^•−^ able to reduce Compound II at a sufficient rate can enhance the removal of peroxynitrite as a result of permanent switching between resting MPO and Compound II (see Figure 1) [74].

## 3. Neutrophils and Myeloperoxidase at Inflammatory Sites

### 3.1. Recruitment of PMNs to Inflamed Sites

Neutrophils circulate in peripheral blood in huge amounts. With their receptors, they are able to sense any changes in properties of the vessel wall that allows these cells to adhere firmly to the endothelium and to permeate into inflamed regions. During and after diapedesis, this directed movement is driven by chemical gradients of cytokines and chemotactic agents such as interleukin-8, interferon γ, complement factors C3a and C5a, fMet-Leu-Phe, and leukotriene B_4_ [75,76,77,78,79].

In resting polymorphonuclear leukocytes, myeloperoxidase is stored in the so-called azurophilic granules. Other types of granules of these cells are secretory, tertiary, and specific granules. These granule types differ in their internal and membranous composition of active agents and dependence on cytoplasmic calcium levels to induce degranulation [80,81]. During the recruitment of neutrophils to inflamed tissues, these cells are step-by-step activated with the release of granule contents into the cell environment or into newly formed phagosomes containing microbes of fungi. The released granule components from secretory, tertiary and, to some extent, specific granules help to digest the surrounding connective tissue to facilitate tissue invasion of these cells that allow a directed movement of invading cells to the inflamed area, where foreign microorganisms are phagocytosed.

### 3.2. Important Components of Azurophilic and Specific Granules of Neutrophils

Components of specific and especially azurophilic granules are mostly involved in deactivation and killing of ingested microbes and fungi. They discharge their contents into the newly formed phagosomes. Azurophilic granules deliver, besides myeloperoxidase, serine proteases, bactericidal/permeability-increasing protein, defensins, lysozyme, azurocidin, and others. The main components of specific granules are lysozyme, lactoferrin, some serine proteinases, histaminase, type IV collagenases, gelatinase, and others [82]. In the formed phagosomes, the ingested microbes and fungi are surrounded by a small fluid volume containing highly concentrated material discharged from fused granules [83].

Granule components exhibit different functions in the formed phagosomes. Major functions of these components in the early phase of phagocytosis are listed in Figure 2. Some of them are known to interact with surface structures of ingested microorganisms, and thus contribute to the inactivation and killing of pathogens. Prominent examples are defensins, bactericidal/permeability-increasing protein, and azurocidin [84,85,86]. Lactoferrin exerts also an antibacterial activity by interaction with outer bacterial membranes. Additionally, this protein sequesters free iron ions [87,88]. Another group of agents has a pH optimum around 8–9 such as the serine proteases elastase, cathepsin G, and proteinase 3 as well as the hydrolase lysozyme [89,90]. Neutrophil collagenase exhibits maximal activities between pH 6 and 9.5 [91]. The pH-dependence of enzymatic activity of PMN gelatinase is more bell-shaped, with a maximum between pH 7.5 and 8 [91]. As shown before, myeloperoxidase needs to be activated by reactive species such as H_2_O_2_, or O_2_^•−^, or peroxynitrite. This enzyme is principally active under alkaline, neutral, and acidic conditions. However, the chlorination and peroxidase activity of MPO is optimal at pH 5–6.

### 3.3. Conditions of Phagosomal Digestion

Components of fused granules participate in the creation of a special milieu in phagosomes, which facilitates the inactivation of microorganisms [83]. At the beginning of phagocytosis, NADPH oxidase is assembled from cytoplasmic and membranous components [92]. This transmembrane protein generates a large amount of superoxide anion radicals (O_2_^•−^) by the oxidation of cytoplasmic NADPH, transfer of electrons through the phagosomal membrane, and one-electron reduction of O_2_ in the phagosomal space. Dismutation of O_2_^•−^ yields hydrogen peroxide and dioxygen. The enhanced formation of O_2_^•−^, opening of ion channels, and altered ion fluxes through the phagosomal membrane contribute to a rapid increase in pH in newly formed phagosomes [83,93].

Originally, it was assumed that the phagosomal pH increases rapidly to about 7.8–8.0 (during the first three minutes), and decreases gradually to 7.0 (after 10–15 min) and later to 6.0 (after about 1 h) [94,95]. The application of a better suitable pH indicator and the removal of azide from the incubation medium revealed a much higher and more long-lasting pH increase to a mean value of 9.0 (in few cases to pH 10.2) in human PMNs incubated with deactivated *Candida albicans* [96,97]. Concomitantly with the rise in phagosomal pH in human PMNs, the cytoplasmic pH drops slightly from 7.56 to a medium value of 7.3 [96].

The significance of NADPH oxidase in an increase in phagosomal pH is demonstrated by the use of the NADPH oxidase inhibitor diphenylene iodonium (DPI) or by the application of NADPH oxidase-deficient neutrophils. Under these conditions, the phagosomal pH drops down to about 6.3 and slightly lower [96,97].

In newly formed phagosomes, an alkaline milieu is important for the rapid and efficient inactivation of phagocytosed microorganisms. Numerous bactericidal constituents of azurophilic and specific granules have a pH optimum around 8–9, such as elastase, cathepsin G, proteinase 3, and lysozyme [89,90]. Neutrophil collagenase and gelatinase also exhibit significant activities in this pH region [91] (Figure 2). Thus, a cocktail of aggressive proteins, bactericidal proteins and reactive species is present in the formed phagosomes. The concerted action of these agents promotes the deactivation, killing, and digestion of the phagocytosed microorganisms.

### 3.4. Potential Role of Myeloperoxidase in Phagosomes

In azurophilic granules of resting neutrophils, MPO as well as other cationic protein components are probably inactivated by sequestration with negatively charged proteoglycans and the presence of low pH [98,99]. These conditions are changed when azurophilic granules discharge their content into the newly formed phagosomes of activated PMNs.

Immediately after phagosome formation, the pH rises to alkaline values as a result of NADPH oxidase activation and compensatory ion fluxes [83,96]. The consumption of H^+^ ions in reactions of the formed O_2_^•−^ is responsible for this pH increase. During the early phase of phagocytosis, discharged MPO can likely act as a weak superoxide dismutase, considerably enhancing the removal of O_2_^•−^. In one dismutase cycle, two protons are consumed.

Moreover, the high yield and stability of O_2_^•−^ promotes also its reaction with nitrogen monoxide (NO), an agent that is freely diffusible. The reaction product of O_2_^•−^ and NO is peroxynitrite (ONOO^−^). The protonated form of ONOO^−^, peroxynitrous acid ONOOH, is known to convert resting MPO directly into Compound II [73]. As O_2_^•−^ efficiently recovers resting MPO from Compound II [66], peroxynitrite removal by MPO can drive a permanent cycling between resting MPO and Compound II. Two protons are consumed during one cycle. The production of peroxynitrite also contributes to microbicidal activity in phagosomes [100].

Superoxide anion radicals also interact with sulfur–iron-clusters under the release of Fe^2+^ [101,102]. This iron can be scavenged by lactoferrin [88].

At alkaline pH values, MPO is far from the optimum for efficient halogenating and peroxidase activities. Under these conditions, MPO is unable to oxidize chloride at sufficient rate. However, it cannot be excluded that hydrogen peroxide favors Compound I formation and the reaction of Compound I with traces of SCN^−^ under formation of ^−^OSCN in the halogenation cycle or the oxidation of O_2_^•−^ to O_2_ in the peroxidase cycle.

Later, the phagosomal pH decreases step by step, reaching neutral values. It remains unsolved whether some phagosomes become leaky during phagocytosis with changes of pH to cytoplasmic level. Under these conditions, the dominance of O_2_^•−^ is diminished, and the halogenating and peroxidase activity of MPO activity rises slowly. It remains unsolved whether acidic pH values can be achieved in phagosomes of neutrophils.

We are far from a thorough understanding of the role of MPO during the early phase of phagocytosis in PMNs. The main reason for this is the lack of sufficient data about MPO reactions in the pH region 8–10, typical of early phagosomal pH values. It is quite evident that under these conditions, most reactions of halogenation and peroxidase cycles do not work or only work to a limited degree. Otherwise, superoxide anion radicals are stable enough due to the low rate of spontaneous dismutation and can drive now O_2_^•−^-dependent reactions of MPO. An enhanced catalase activity of MPO was also proposed in the pH region 9–10 [96]. However, a mechanism for this reactivity was not given. Compound III is indeed involved in a catalase-like removal of H_2_O_2_ by the sequential conversion: Compound III→ferrous MPO→Compound II→Compound III [103]. As the H_2_O_2_-driven conversion of Compound II to Compound III is the rate-determining step, this cycle can explain the observed accumulation of Compound II at alkaline pH [104].

The application of different inhibitors revealed cooperative effects of superoxide anion radicals and a myeloperoxidase-dependent pathway in the killing of *Staphylococcus aureus* by neutrophils [105]. Despite the strong alkalization of early phagosomes in human neutrophils [96], the authors excluded a sufficient participation of MPO in these reactions, as the MPO inhibitors 4-aminobenzoic acid hydrazide (4-ABAH) and KCN failed to affect the phagosomal pH. The inhibitor 4-ABAH is known to interact with Compound I [106], which is apparently not involved in O_2_^•−^-driven MPO reactions under alkaline conditions. Hydrocyanic acid (HCN) forms a complex with resting MPO, whereby HCN is deprotonated by the interaction with distal histidine [107]. This binding is optimal between pH values 5 and 8. At pH > 8, the apparent second-order rate constant for the complex formation between MPO and HCN decreases with increasing pH due to the *p*K_a_ value of 9.2 for HCN [107]. Considering the additional presence of O_2_^•−^, which can compete with HCN for ferric MPO, it is questionable whether KCN is able to inhibit MPO at alkaline pH values in the phagosome. Azide, another MPO inhibitor, caused a significant decrease in phagosomal pH [96] indicating an involvement of MPO in phagosome alkalinization.

In vitro experiments demonstrate the killing of microbes by the MPO-H_2_O_2_-halide system and inhibition of killing by MPO inhibitors or by the application of neutrophils from MPO-deficient individuals [6,108,109,110,111]. From these data, it was concluded that halogenating MPO is critical for the inactivation of microbes. These experiments are usually performed at a low protein load, at neutral pH values, and in the absence of competing substrates, conditions that are far from the actual situation in phagosomes. Product analysis of phagocytosed material of PMNs revealed that most other released granule components were halogenated but not constituents from ingested microbes [112,113]. Under these conditions, bacteria are apparently not killed by halogenated MPO products.

Other data support an involvement of the halogenating activity of MPO in the termination of PMN serine proteases in phagosomes and protection of pericellular tissues from unwanted reactions. For example, the MPO product HOCl is known to inactivate elastase, cathepsin G, proteinase 3, and matrix metalloproteinase 9 [114].

In summary, in the early phase of phagocytosis in neutrophils, MPO apparently helps to intensify and prolong the duration of an alkaline milieu by activities mainly based on the consumption of O_2_^•−^. Later, MPO can probably contribute with its halogenation and peroxidase activity to the termination of phagocytotic activities and protection of surrounding tissues from uncontrolled proteolysis.

### 3.5. Redundancy in Deactivation and Killing Pathways and MPO Deficiency

In activated neutrophils, a broad range of active agents ensures protection against different types of microbes and fungi. There are multiple deactivation and killing pathways and a redundancy exists between these pathways.

For example, in humans, total and subtotal MPO-deficiency occurs with an abundance of 1 case on 2000–4000 individuals [115,116]. As opposed to patients with an NADPH oxidase deficiency, which develop chronic granulomatous disease and suffer from multiple inflammatory events [117], MPO-deficient persons live normally. Only in few cases were persistent infections with *Candida albicans* reported [118,119,120].

Unlike humans, serious complications were reported in investigations of different disease models using MPO-knockout mice [111,114,121,122,123]. A careful analysis revealed a protective role of MPO in pathologies characterized by the infiltration of T-cells. MPO oxidants are involved in the suppression of lymphocyte functions. Otherwise, in disease models, where an innate immune response dominates exclusively, MPO knockout protects tissue from damage.

In MPO-knockout mice, no MPO protein is present. Human MPO deficiency is mainly assessed by diminished or missing peroxidase activity [116], a property that is more pronounced under slightly acidic conditions and not in the early phase of phagocytosis. It remains unsolved whether MPO from MPO-deficient patients exhibits a superoxide dismutase-like activity, which can be involved in the alkalization of newly formed phagosomes.

Moreover, human and murine neutrophils differ considerably in their properties [124]. Mouse neutrophils do not contain any defensins [125], small cationic proteins that induce defects in microbial membranes [126]. In azurophilic granules of human PMNs, four types of defensins are present. Other differences concern the lower expression of bactericidal/permeability-increasing protein, lysozyme, and β-glucuronidase in mouse neutrophils [127,128,129]. The same holds for MPO, with a level of 10–20% in murine compared to human cells [127,128]. A further difference concerns the degree and duration of the pH increase during the early phase in phagosomes. In murine neutrophils incubated with *Candida albicans*, the pH reaches with 8.5 a lower mean maximum value and decreases faster than in human cells [96]. It remains unknown to what extent the difference in MPO expression in both species contributes to the more pronounced increase in pH in phagosomes of human neutrophils. A higher amount of MPO in human cells can more strongly enhance the dismutation of O_2_^•−^ and can utilize more H_2_O_2_.

These data support a higher degree of redundancy in deactivation and killing pathways in human neutrophils than in murine ones.

### 3.6. Cell Death of Neutrophils and Formation of Extracellular Traps

Activated neutrophils are known to undergo apoptosis [130,131]. Typical features of apoptotic cells are the appearance of phosphatidylserine epitopes on the cell surface by concurrent intactness of the plasma membrane. In necrotic cells, the permeability of the plasma membrane increases. At inflammatory sites, apoptotic and necrotic PMNs are recognized and ingested by macrophages that are attracted to these sites, usually time-delayed to PMNs [132,133]. In both apoptotic and necrotic PMNs, catalytically active myeloperoxidase has been found attached to phosphatidylserine epitopes [134].

In dying PMNs, so-called neutrophil extracellular traps (NETs) are formed and released. These traps represent a network of extracellular DNA-derived fibers, to which MPO, elastase, cathepsin G, gelatinase, lactoferrin, calprotectin and some others are attached [135,136]. Extruded traps can bind and kill microbes independent of phagocytosis [135,137]. It is discussed that NETs are important in host defense against fungal pathogens that are difficult to phagocytose [136,138]. In blood, NETs are able to promote coagulation, vascular occlusion, thrombosis, sequestration of circulating tumor cells, and metastasis [139,140,141]. In addition, both pro- and anti-inflammatory activities of NETs are reported [15]. NETs are internalized and degraded by macrophages [15].

Myeloperoxidase is required for NETs formation, as PMNs from patients completely deficient in MPO cannot form traps [142]. Hydrogen peroxide is known to enhance the enzymatic activity of MPO in traps and the ability of NETs to kill microbes [143].

### 3.7. Frustrated Phagocytosis

Besides necrotic cell death, aggressive agents from activated PMNs can also be released into the surrounding medium as a result of an incomplete, the so-called frustrated, phagocytosis [144,145]. In this case, parts of specific and azurophilic granules fuse with the plasma membrane and discharge their content into external space. This may occur in premature or overstimulated neutrophils, at extended inflammatory loci, or as a result of a high load of pathogens. In this way, MPO, serine proteases, and other granule constituents are released from neutrophils, and can interact with host cells and tissues.

### 3.8. Degradation of Ingested Material by Macrophages

At inflammatory sites, tissue-resident and monocyte-derived macrophages engulf and digest waste material and undergoing cells including apoptotic and necrotic neutrophils. It has been assumed that macrophages can modulate their activity depending on the dominance of apoptotic and necrotic cell material [146,147,148,149]. An inflammation persists as long as constituents released from necrotic cells contribute to further cell and tissue damage. Under these conditions, activated macrophages mainly release proinflammatory cytokines. For example, some components from necrotic PMNs such as elastase, cathepsin G, and heat shock proteins are known to modify surface molecules of adjacent macrophages, and thus induce pro-inflammatory pathways in the latter ones. A switch to the resolution of inflammation and the release of mainly anti-inflammatory cytokines occurs when macrophages digest predominantly apoptotic cell material [131,150,151,152,153].

An important aspect of the interaction between apoptotic PMNs and macrophages is the rapid recognition of apoptotic cells by macrophages. Their delayed phagocytosis by macrophages can result in secondary necrotic processes [131,154,155].

Interestingly, in classically activated macrophages (type 1 macrophages) from human origin, the phagosomal pH rises to about 8.5 within five minutes, displays some oscillations of alkalinization, and remains nearly unchanged or returns to neutral values during the next 25 min [97,156]. This pH increase depends on NADPH oxidase activity, like in phagocytosing neutrophils. Contrary to this, alternatively activated macrophages (type 2 macrophages) monotonically acidify their phagosomes to pH 5.0 within 10 min [156] or to pH 5.5 within 30 min without any participation of NADPH oxidase [97]. Intriguingly, in murine type 1 macrophages, which express unlike human cells nitric oxide synthase [157,158], the phagosomes are also acidified [159,160]. Apparently, the very rapid reaction between NO and O_2_^•−^ [71,72] diminishes the availability of O_2_^•−^ in these cells and limits, thus, the consumption of H^+^ by O_2_^•−^-driven reactions.

The presence of MPO on surface epitopes of non-vital neutrophils and other cells implies the question of whether attached MPO plays a role following phagocytosis of undergoing cells by macrophages. At present, a clear answer on this subject cannot be given. Probably, MPO can play a similar role in alkalinization of the phagosome in classically activated macrophages, as assumed in phagocytosing neutrophils. The strong pH decrease in phagosomes of alternatively activated macrophages would, however, favor better conditions for the chlorinating and peroxidase activity of MPO.

## 4. Involvement of Myeloperoxidase in Disease Progression

### 4.1. The Fate of Myeloperoxidase at Inflammatory Sites

Myeloperoxidase can be released from PMNs at inflammatory sites as a result of incomplete phagocytosis, cell necrosis, or as component of NETs. The cationic nature of MPO favors an interaction with numerous negatively charged serum proteins, endothelial cells, and components of the extracellular matrix. The activation of MPO by hydrogen peroxide causes halogenation and oxidation reactions in critical residues of the attached targets. On the basis of these interrelations, an involvement of MPO in several pathologies is discussed [9,161,162].

### 4.2. Important Binding Sites for Myeloperoxidase

Myeloperoxidase is known to interact with numerous plasma proteins. Critical residues in albumin [54,163,164,165], α_1_-antiproteinase (also known as α_1_-antitrypsin) [166,167,168,169], apolipoprotein A_1_ [170,171], and soluble plasma fibronectin [172,173] are modified by the interaction of MPO with these targets.

Myeloperoxidase binds to heparin/heparan-containing epitopes of endothelial membranes [174]. Complexes between albumin and MPO are also known to interact with the endothelium [164]. Further, MPO is internalized by endothelial cells, rapidly transcytoses the intact endothelium, and associates closely with fibronectin [175]. In an inflammatory model, MPO diminishes the bioavailability of NO and impairs the NO-dependent vessel relaxation [176].

In the extracellular matrix and coronary smooth muscle cells, MPO is known to bind to collagen IV and fibronectin [177,178]. Other targets of MPO are perlecan [179], laminin [180], and glycocalyx glycosaminoglycans [181].

### 4.3. The Protective Role of Ceruloplasmin

However, the incubation of purified MPO with plasma revealed ceruloplasmin as the major binding target for MPO [182,183]. In these experiments, co-elution between MPO and complement C3 was also observed. All other reported in vitro associations of MPO with serum proteins have likely only minor physiological significance.

The copper-containing ceruloplasmin is a serum protein with multiple anti-inflammatory and antioxidant functions. It oxidizes Fe^2+^ and Cu^+^ [184]. Ceruloplasmin forms high-affinity complexes with myeloperoxidase and other cationic proteins of neutrophils such as lactoferrin and serine proteases [185,186]. Its complex with lactoferrin facilitates the binding of the resulting Fe^3+^ and the transfer to the iron-binding protein transferrin [186]. A superoxide dismutase activity of ceruloplasmin is also reported [187].

The complex formation between MPO and ceruloplasmin strongly inhibits the halogenating and peroxidase activity of MPO [188,189,190]. In the complex between MPO and ceruloplasmin, MPO is converted to Compound II and held in this inactive form until ceruloplasmin dissociates from MPO [183]. Inhibition of MPO activities is favored by a loop of the ceruloplasmin chain that penetrates into the substrate channel and blocks the access to the active site of MPO [191,192]. Partial proteolysis of ceruloplasmin by elastase, plasmin, or trypsin dampens this inhibition [190]. Antibodies against MPO can prevent the binding between MPO and ceruloplasmin by steric hindrance. Hence, this interaction with antibodies favors the presence of active MPO in patients with renal vasculitis [193].

### 4.4. Myeloperoxidase in Atherosclerotic Plaques

Atherosclerotic plaques limit vessel lumen, disturb blood circulation, and cause serious health problems for the affected patients. In plaque formation, lipoproteins, endothelial cells, immune cells, smooth muscle cells, and others are involved [194,195,196,197,198]. Oxidative modifications to lipoproteins, extracellular matrix components, and cell surface elements also contribute to the formation of atherosclerotic lesions [178,199,200]. Despite intense research, it remains puzzling how atherosclerosis is initiated in vivo and how MPO participates in this pathogenesis.

In atherosclerotic lesions, active MPO was detected [201,202]. Chlorinated and nitrated products such as 3-chlorotyrosine, 3-chlorouracil and 3-nitrotyrosine were found in plaque proteins as well [203,204]. While chlorinated products are specific to MPO and associated with the MPO-mediated formation of HOCl, nitration can also result from peroxynitrite-driven reactions. In atherosclerotic lesions, numerous components of the extracellular matrix are oxidatively modified, such as fibronectin, laminin, and others. Again, the involvement of MPO is discussed [177,178,200]. Moreover, there is an increased affinity of MPO to oxidized material in the extracellular matrix [178]. This positive feedback can further promote oxidative matrix modifications in developing atherosclerotic lesions.

It remains unclear whether the interaction of MPO with lipoproteins and endothelium components is the initial reason for plaque formation, or whether the presence of MPO and MPO products in plaques results from subsequent processes like the infiltration of PMNs into inflamed areas.

In addition, several further MPO-related mechanisms are assumed to play a potential role in plaque development. Myeloperoxidase-derived HOCl is known to oxidize low-density lipoproteins and convert them into a pro-atherogenic species [205,206]. The interaction of MPO with apolipoprotein A1, the major protein in high-density lipoproteins (HDLs), limits the ability of HDLs to remove cholesterol from lipid-laden cells [170,171,207,208]. The formation of cyanate as a result of SCN^−^ or cyanide oxidation by MPO leads to the carbamylation of lysine residues in proteins, a process that has been associated with foam cell formation [209,210]. Myeloperoxidase is involved by several routes in protein carbamylation in atherosclerotic lesions [210]. Enhanced MPO plasma levels probably decrease the bioavailability of nitric oxide, and can thus promote endothelial dysfunction [211,212]. Another consequence of MPO activity is the enhanced activity of matrix metalloproteinases [213,214]. It should be noted that there are other pathways independent of MPO leading to LDL oxidation, diminished bioavailability of NO, disturbed endothelial dysfunctions, or protein carbamylation.

### 4.5. Myeloperoxidase and Cardiovascular Diseases

There are several reports specifying a relationship between enhanced MPO plasma levels and the development of cardio-vascular problems [215,216,217]. Again, the question arises of whether the enhanced MPO level is the origin of these problems or the consequence of the additional recruitment of neutrophils to inflamed endothelium in these patients.

Rupture of atherosclerotic plaques is a major reason for acute cardiovascular disease [218]. Among critical factors in plaque formation, the infiltration of neutrophils and macrophages contributes to weakening of the fibrous cap, a process that precedes plaque rupture [200]. Determination of MPO plasma levels has been used as a marker for the predictive outcome of cardiovascular problems. The majority of studies demonstrated a relationship between elevated plasma MPO levels and the presence of coronary artery disease [200].

### 4.6. Vasculitis Induced by Antineutrophil Cytoplasmic Antibodies

Antineutrophil cytoplasmic antibodies (ANCA) are involved in ANCA-associated vasculitis, which represents a small vessel vasculitis. There are mainly antibodies against myeloperoxidase and, to a lesser extent, against proteinase 3 [219]. Both proteins are present in a high load in neutrophils and are known to be attached to the surface of dying neutrophils [134,220]. It cannot be excluded that these proteins are also be found on surface areas of other undergoing cells at inflammatory sites and, hence, are presented as antigens by dendritic cells.

The development of MPO-ANCA and other ANCA is poorly understood [221]. Among the postulated mechanisms of ANCA generation are defective apoptosis of neutrophils [222] and the hypothesis of molecular mimicry [223].

Once formed, ANCA can recognize over-activated neutrophils, which are attached to inflamed vessel wall areas. These attacks further promote the inflammatory process, leading to a necrotizing vasculitis. Glomerulonephritis, vasculitis of the upper and lower respiratory tract, and diabetic retinopathy are prominent forms of ANCA-associated vasculitis [224,225,226].

In the pathogenesis of different types of vasculitis, a relationship between MPO and α_1_-antiproteinase is discussed [193,227]. Myeloperoxidase products oxidize critical methionine residues in this antiproteinase [166,167]. As a result, the ability of α_1_-antiproteinase is limited to inactivate elastase and proteinase 3 at inflammatory sites and a partial protease-mediated hydrolysis of ceruloplasmin may occur. Partially proteolyzed ceruloplasmin is unable to inhibit MPO [190]. Collectively, the imbalance between the protease–antiproteinase activities can favor a prolonged activity of myeloperoxidase and proteinase 3 and the formation of antibodies against both agents.

### 4.7. Involvement of Myeloperoxidase in Other Disease Scenarios

In many diseases, the infiltration of neutrophils into inflamed tissue regions is closely associated with increased MPO levels. Active MPO was detected in tissue sections of patients with pancreatitis [228], in periodontitis [229], and in synovial fluid of patients with rheumatoid arthritis [230]. The involvement of MPO is also discussed in the pathogenesis of Alzheimer’s disease [231,232], Parkinson’s disease [233,234], and multiple sclerosis [11,235]. Further examples of the potential involvement of MPO in disease pathogenesis are obesity [236], sinusitis [237], cystic fibrosis [238], inflammatory bowel disease [239,240], renal [241,242], and liver diseases [243].

## 5. Chronicity of Inflammatory States

### 5.1. Chronic Inflammatory Processes

Many disease scenarios are accompanied by chronic inflammatory processes, in which the inflammation is only insufficiently terminated. In inflamed tissues, there is a mixture of unperturbed tissue regions with necrotic areas, and regions where a de novo reconstitution of tissues occurs. Generally, it is impossible to give a clear description of chronically affected tissues as the appearance of these regions varies enormously. There are also flashes of the active inflammation with silent phases.

In the termination of inflammation, a switch occurs from a pro-inflammatory phase to resolution of inflammation followed by restoration of tissue homeostasis [244,245,246,247]. In chronic inflammations, damage-associated molecular patterns (DAMPs) released from necrotic cells can repeatedly foment the inflammatory process [248]. A crucial aspect for chronicity is the balance between the damage to host’s tissues by agents from activated immune cells and undergoing tissue cells and the ability of host’s tissues to resist and inactivate these destructive agents [1]. There is a wide range of variability of protective mechanisms from one patient to another. The low-tissue capacity of antagonizing agents generally prolongs inflammation, favors its chronic conversion, and provides the basis for disease progression. Thus, the preponderance of damaging over protective mechanisms disturbs the termination of inflammation and favors long-lasting inflammatory events.

### 5.2. Protection of Surrounding Media Against Damaging Agents

In human blood and tissues, numerous mechanisms and strategies exist to protect biological material at inflammatory sites from damage caused by destroying agents released from activated neutrophils, other immune cells, and damaged tissue cells. Known examples for the latter agents are free hemoglobin, and free myoglobin, which are released from red blood cells during intravascular hemolysis [249,250], and muscle cells as a result of rhabdomyolysis [251,252], respectively. Both free hemoglobin and free myoglobin are the source of the very cytotoxic agent ferric protoporphyrin IX (free heme) [253,254,255,256].

Intact cells and tissues are generally protected by the presence of molecules, which deactivate cytotoxic agents by binding, inactivation, or degradation. Examples of these protective interactions are given in Table 3. It should be noted that these mechanisms are mostly directed against frequently occurring damaging agents.

Some of the listed protecting species are acute phase proteins (such as ceruloplasmin, α_1_-antiproteinase, α_1_-antichymotrypsin, and haptoglobin), which can be up-regulated to a certain degree during the course of inflammation [270,271]. Otherwise, at prolonged and severe inflammatory events, a decrease in or exhaustion of distinct acute phase proteins and other protective agents can be observed.

Released myeloperoxidase is inactivated by complex formation with ceruloplasmin [188,189,190], which is a late acute-phase protein and reaches its maximal plasma level after most other acute-phase proteins [272]. The question arises of whether this interaction with ceruloplasmin is sufficient to inactivate all released MPO under severe inflammatory conditions.

As shown in Table 3, protective mechanisms are very broad-ranged and comprise numerous anti-proteinases, control over superoxide anion radicals, hydrogen peroxide, and free metal ions, the presence of lipid- and water-soluble antioxidants and agents, to detoxify free heme and free heme proteins. These immediately acting, ready-to-use protective mechanisms are completed by inducible processes that enhance the protective power of cells, and by formation of antibodies against antigens.

Of the antagonizing agents against neutrophil products, α_1_-antiproteinase deserves special attention. This acute phase protein is produced in the liver and protects the lungs and other tissues from neutrophil elastase [273,274]. Individuals with α_1_-antiproteinase deficiency may develop chronic obstructive pulmonary disease, chronic liver disease, skin inflammations, ANCA-related vasculitis, glomerulonephritis, and Bowel disease [275,276,277]. The protease-anti-proteinase imbalance can also affect the ability of ceruloplasmin to bind and inactivate MPO [190].

### 5.3. Immunosuppression

During the resolution of inflammation, transient immunosuppression is important to allow final apoptosis of immune cells, to restore the normal tissue homeostasis, to terminate all inflammatory processes, and to induce, if necessary, de novo formation of tissues [245,278]. Many chronic disease processes are accompanied by pronounced immunocompromised states of patients. This mostly concerns persons of advanced age, who suffer from different comorbidities [279]. However, long-lasting immunosuppression depresses general immune functions, and favors the appearance of opportunistic infections [280,281].

### 5.4. Sepsis

Strong deviations of normal tissue homeostasis can result in sepsis. According to the last consensus definition in 2016 [282], sepsis is, at present, defined as life-threatening organ dysfunction caused by a dysregulated host response to infection. Septic shock is determined as a subset of sepsis, in which the underlying circulatory and cellular/metabolic abnormalities are profound enough to substantially increase mortality [282]. Sepsis is characterized by malfunction of one or several organs such as liver, kidneys, or lungs. It occurs most frequently in immunocompromised individuals [283,284].

Immunosuppression impacts PMN functions in septic patients. These cells show a delayed apoptosis and diminished chemotactic mobility [285,286]. With these properties, PMNs can contribute to tissue damage distant from inflammatory and infection sites [287,288]. In addition, an increased percentage of immature neutrophils is found in septic patients [286].

In sepsis, tissue-damaging processes can vary widely and depend largely on the state of individual protective mechanisms [289,290]. Besides invading microorganisms, products of neutrophils including MPO, and products of other immune and tissue cells, can be involved in cell and tissue damage in septic patients. Furthermore, antigens and DAMPs released form necrotic tissue cells also play a role in the recruitment and activation of immune cells. The considerable decline or exhaustion of some protective mechanisms also worsens the recovery of unperturbed tissue homeostasis.

In septic shock patients, plasma levels of MPO–DNA complexes, which result from neutrophil extracellular traps, are significantly enhanced in contrast to healthy volunteers and closely related to the severity of organ dysfunction [291]. Patients with sepsis or with septic shock show higher plasma levels of MPO in comparison to patients with systemic inflammatory response syndrome without infection [292]. In addition, an association between higher MPO levels and increased mortality was found [292]. Enhanced MPO values were also detected in other studies with septic patients [293,294].

## 6. Conclusions

The fate of myeloperoxidase reflects the general role of neutrophils in immune response. On the one hand, neutrophils and neutrophil constituents are highly essential to successfully combat foreign bacteria and other pathogens. In the phagosomes of PMNs, myeloperoxidase is involved in the creation and maintenance of an alkaline milieu, which is optimal for the activity of serine proteases and other granule components in the deactivation and killing of microbes. Halogenation and peroxidative activities of MPO are apparently involved in later phases of phagocytosis of pathogens and can play a role in the control and termination of phagocytic activities in neutrophils, and maybe also in macrophages, after the ingestion of undergoing PMNs. In addition, MPO is a mandatory element for the formation of neutrophil extracellular traps.

On the other hand, the release of PMNs constituents during frustrated phagocytosis or from necrotic PMNs can damage intact host tissues. Myeloperoxidase as a major constituent of PMNs contributing to this damage by its close association with negatively charged components of plasma and extracellular matrix and by the chemical modification of these targets. The acute phase protein ceruloplasmin binds and inactivates MPO. The ability of MPO to damage molecules and tissues can be enhanced at inflammatory sites by the massive release of MPO from PMNs and the imbalance between proteases and anti-proteinases. In several disease scenarios and sepsis, the involvement of MPO in the pathological process is discussed. It remains unknown whether MPO contributes to basic mechanisms of disease induction or whether the increased MPO level in affected tissues results from downstream effects due to the recruitment of neutrophils to inflammatory sites.

In sum, MPO exhibits, like neutrophils and many other components of the immune system, both a protective and harmful role in the maintenance and disturbance of tissue homeostasis.

## Figures and Tables

**Figure 1 ijms-21-08057-f001:**
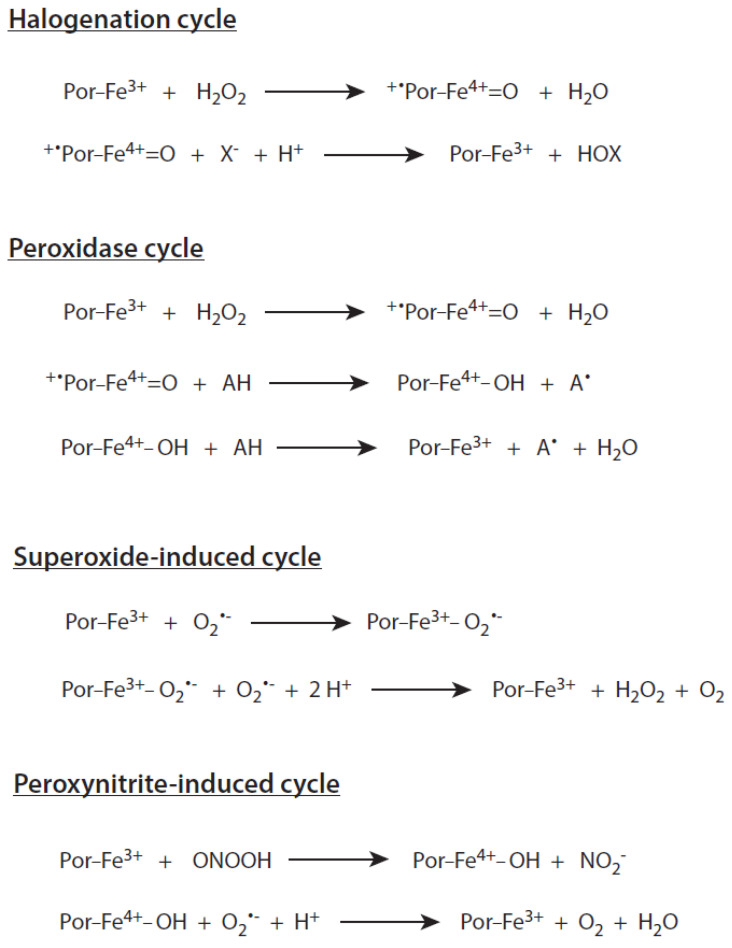
Major catalytic cycles of myeloperoxidase. Further explanations are given in the text. The heme states of MPO are denominated as in Table 1. X^−^ stands for Cl^−^, Br^−^, I^−^, and SCN^−^. HOX is the corresponding (pseudo)hypohalous acid. AH is an oxidizable substrate, and A^•^ the resulting substrate radical.

**Figure 2 ijms-21-08057-f002:**
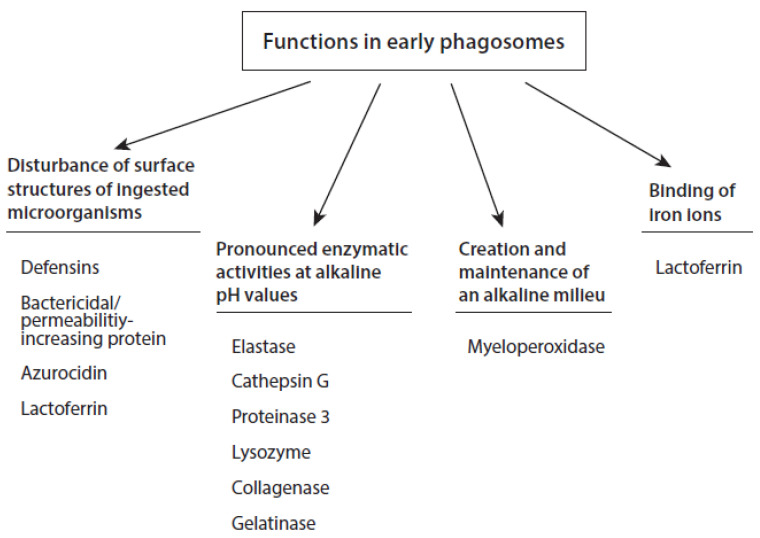
Major functions of granule constituents of PMNs in early phagosomes.

**Table 1 ijms-21-08057-t001:** Important heme states of myeloperoxidase.

Heme State of MPO	Short Denomination	Formal Oxidation State Versus Resting MPO
resting MPO	Por−Fe^3+ a^	
Compound I	^•+^Por−Fe^4+^=O	+2
Compound II	Por−Fe^4+^−OH	+1
Compound III	Por−Fe^3+^−O_2_^•−^	0

^a^ Por denotes the porphyrin ring.

**Table 2 ijms-21-08057-t002:** Standard reduction potentials (*E*’°) for the interconversion between heme states of human myeloperoxidase.

Redox Couple	*E*’° (at pH 7)	Number of Transferred Electrons	References
Compound I/resting MPO	1.16 V	2	[24]
Compound I/Compound II	1.35 V	1	[25]
Compound II/resting MPO	0.97 V	1	[25]

**Table 3 ijms-21-08057-t003:** Immediately acting protective mechanisms against frequently occurring damaging agents.

Potentially Damaging Agents	Antagonizing Principle	References
Myeloperoxidase	Ceruloplasmin	[182,183]
Proteases from PMNs, mast cells and others	Anti-proteinases such as α_1_-antiproteinase, α_1_-antichymotrypsin, secretory leukocyte protease inhibitor, elafin, α_2_-macroglobulin	[90]
Superoxide anion radicals	Superoxide dismutases, cytochrome c, ceruloplasmin	[187,257,258,259]
Hydrogen peroxide	Glutathione peroxidases, peroxiredoxins, catalase	[260,261,262]
Free metal ions	Ceruloplasmin, chelators, lactoferrin, ferritin	[88,184,263,264]
Free hemoglobin, free myoglobin	Haptoglobin	[265]
Free heme	Hemopexin	[265]
Lipid-based oxidants	Lipid-soluble antioxidants such as tocopherols, carotenoids, ubiquinol	[266,267]
Water-based oxidants	Ascorbic acid, urate	[268,269]

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
