# Peer review of "The Dual Role of Myeloperoxidase in Immune Response"

_ijms, 2020, doi:10.3390/ijms21218057_

Round 1

Reviewer 1 Report

The author has submitted a review on the 'dual role of mpo in immune response'

The strength of this manuscript is the detailed description of the overview of myeloperoxidase properties, role of MPO in the phagosome the overall illustration of MPO and the alkaline milieu.

The weakness of the manuscript are some of the subsections where MPO is described in the setting of diseases. This has been a recently covered well in recent reviews and I suggest the author consults the latest literature as a guide in this instance. for example doi: 10.3390/medsci6020033 

The review will benefit significantly from a table of contents appearing after the abstract. 

To improve the flow of the manuscript, I recommend:

  • combining section 3.6 and 3.8. It is difficult to separate neutrophil apoptosis and neutrophil extracellular traps as this is the specialised form of neutrophil cell death and thus make sense to appear in the same subsection.
  • section 4.1, line 405 the author states that will focus on the potential role of MPO in cardiovascular diseases, howevert the author also discussesMPO in the context of vasculitis with no particular focus on cardiovascular disease.I recommend the authors remove this line.
  • in section 4.7, the author briefly discusses the correlation between myeloperoxidase and various diseases. The authors should include inflammatory bowel diseases where the contribution of myeloperoxidase is well supported in the literature..
  • section 5.3 and section 5.4 do not provide sufficient evidence or review on the impact of MPO on sepsis and either should be removed entirely or significantly rewritten with further evidence from the literature on the role of MPO in this disease state. Sepsis results in a massively mounted immune response in an acute setting and unsurprisingly there would be considerable levels of plasma MPO  detected. However, more evidence  is required to properly delineate a specific role for MPO during sepsis than is provided in this review.
  • subheading for section 4.2 seems inappropriate. is this subsection intended to describe oxidative targets for myeloperoxidase or potential sites where myeloperoxidase is bound (i.e. complexes)?this seems the latter is true and the subheading should reflect this.

Author Response

Thank you very much for your review. The following changes has been performed according to your suggestions.

The reference doi: 10.3390/medsci6020033 and two additional references are cited now in section 4.1.

Sections 3.6 to 3.9 have been reorganized in the revised version of this manuscript. In section 3.6, which is entitled now Cell death of neutrophils and formation of extracellular traps, the first paragraph of section 3.6 and the whole section 3.8 are combined. The former section 3.9 (Degradation of ingested material by macrophages) is now section 3.8. The second and third paragraphs of section 3.6 are included in the novel section 3.8.

In section 4.1, the last sentence has been omitted.

Inflammatory bowel disease is included with two references in section 4.7.

I extended the last paragraph in section 5.4 (Sepsis). Now data from three further references are included, which show enhanced plasma levels of MPO in septic patients.

I changed the subheading of section 4.2 into Important binding sites for myeloperoxidase.

Reviewer 2 Report

The article entitled "The Dual Role of Myeloperoxidase in Immune Response" by Dr. Arnhold is a thorough review regarding the broad functions of myeloperoxidase that includes both historical references and the latest findings. This article has been written with a vast amount of knowledge and a detailed and logical structure by a major figure in the field, and it can be said that it is impeccable. The only point I would like to point out is that the format in Table 3 seems to be slightly broken (maybe the second column is spanning multiple rows, and the boundary is hard to tell). Personally, I am interested in whether it is possible to control inflammation and diseases such as Alzheimer's and Parkinson's by controlling the function of MPO.

Author Response

Thank you very much for your review. The following changes have been performed according to your suggestion.

In Table 3, the third column has been omitted for better clarity of presentation. The information about acute phase proteins contained in this column is now included in the main text (lines 537 and 538).

In section 4.7, I mentioned the potential involvement of MPO in the pathogenesis of Alzheimer's and Parkinson's diseases. The underlying mechanism is the infiltration of neutrophils and other immune cells into inflamed areas, which result from defective tissue regions, and in case of neurodegenerative diseases from undergoing neurons. Of course, an anti-inflammatory treatment can dampen the impact of immune cells on tissue damage, but does not eliminate the original molecular reasons leading to defective tissue regions. Personally, I'm very skeptical does the inhibition of MPO functions significantly contribute to control the inflammatory process. Tissue damage initiated by immune cell activities is very broad-ranged and includes besides MPO many other damaging agents. In addition, and this is a great challenge for therapeutic approaches, tissues of patients differ considerably in the ability of resist and inactivate destructive agents.

Reviewer 3 Report

In my opinion, the author has revised and discussed some recent works concerning the fat of neutrophils- myeloperoxidase in pathogenesis mechanisms. The proposed model for the role of MPO and neutrophils in immune response is only partial consistent with in vitro/experimental results and clinical observations.

I suggest to added and discuss other important aspacts (i.e., generation of MPO-ANCA) and publications in this area.

Author Response

Thank you very much for your review. The following changes has been performed according to your suggestions.

Section 4.6 is already devoted to antineutrophil cytoplasmic antibodies. I added a short paragraph about the generation of MPO-ANCA.